# Small molecule inhibition of lysine-specific demethylase 1 (LSD1) and histone deacetylase (HDAC) alone and in combination in Ewing sarcoma cell lines

**Darcy Welch**[1,2☯], **Elliot Kahen**[1,2☯], **Brooke Fridley**[3‡], **Andrew S. Brohl**[4,5‡], **Christopher L. Cubitt**[2‡], **Damon R. Reed**[1,4,5,6] *

**1** Sunshine Lab, H. Lee Moffitt Cancer Center and Research Institute, Tampa, Florida, United States of America, **2** Translational Research Core, H. Lee Moffitt Cancer Center and Research Institute, Tampa, Florida, United States of America, **3** Department of Biostatistics, H. Lee Moffitt Cancer Center and Research Institute, Tampa, Florida, United States of America, **4** Sarcoma Department, H. Lee Moffitt Cancer Center and Research Institute, Tampa, Florida, United States of America, **5** Chemical Biology and Molecular Medicine Program, H. Lee Moffitt Cancer Center and Research Institute, Tampa, Florida, United States of America, **6** Adolescent and Young Adult Program, H. Lee Moffitt Cancer Center and Research Institute, Tampa, Florida, United States of America

☯ These authors contributed equally to this work.
‡ These authors also contributed equally to this work.
* damon.reed@moffitt.org

**Data Availability Statement:** All relevant data are within the manuscript and its Supporting Information files.

## Abstract

Ewing Sarcoma (ES) is characterized by recurrent translocations between *EWSR*1 and members of the ETS family of transcription factors. The transcriptional activity of the fusion oncoprotein is dependent on interaction with the nucleosome remodeling and deactylase (NuRD) co-repressor complex. While inhibitors of both histone deacetylase (HDAC) and lysine-specific demethylase-1 (LSD1) subunits of the NuRD complex demonstrate single agent activity in preclinical models, combination strategies have not been investigated. We selected 7 clinically utilized chemotherapy agents, or active metabolites thereof, for experimentation: doxorubicin, cyclophosphamide, vincristine, etoposide and irinotecan as well as the HDAC inhibitor romidepsin and the reversible LSD1 inhibitor SP2509. All agents were tested at clinically achievable concentrations in 4 ES cell lines. All possible 2 drug combinations were then tested for potential synergy. Order of addition of second-line conventional combination therapy agents was tested with the addition of SP2509. In two drug experiments, synergy was observed with several combinations, including when SP2509 was paired with topoisomerase inhibitors or romidepsin. Addition of SP2509 after treatment with second-line combination therapy agents enhanced treatment effect. Our findings suggest promising combination treatment strategies that utilize epigenetic agents in ES.

## Introduction

Ewing sarcoma (ES) is the second most common bone sarcoma affecting children and adolescents. Despite advancements in treatment leading to improved outcomes for localized disease

**Funding:** This study was generously supported by the National Pediatric Cancer Foundation (www.nationalpcf.org). This work has been supported in part by the Translational Research Core at the H. Lee Moffitt Cancer Center & Research Institute, a NCI designated Comprehensive Cancer Center (P30-CA076292). The funders had no role in study design, data collection and analysis, decision to publish, or preparation of the manuscript.

**Competing interests:** The authors have declared that no competing interests exist.

**Abbreviations:** LSD1, lysine-specific demethylase 1; ES, Ewing sarcoma; NuRD, nucleosome remodeling and deacetylase co-repressor complex; HDAC, histone deacetylase; 4HC, 4-hydroxycyclophosphamide; FA, fraction affected; FAD, flavin adenine dinucleotide; CI, combination index; $IC_{50}$, half maximal inhibitory concentrations; Cmax, maximal plasma concentrations; AUC, area under the curve; MTIC, (5E)-5-(methylaminohydrazinylidene)imidazole-4-carboxamide.

over time, prognoses remain poor for patients with recurrent or metastatic disease [1]. ES is characterized by translocations involving members of the *ETS* family of transcription factors, most commonly t(11;22)(q24;q12) between the amino terminus of the *EWSR1* gene and the carboxy terminus of the *FLI1* gene, occurring in 85–90% of cases [2, 3]. Efforts to directly target the translocation were reenergized by publication of the ES genomic landscape by three groups demonstrating this alteration to be the only sufficiently recurring change across tumor samples [4–6].

The fusion oncoprotein EWS-FLI1 is considered a transcriptional activator in ES which is required for its oncogeneic activity [3]. A proposed method of targeting the function of the fusion protein is by inhibiting other proteins that may assemble into functional complexes with EWS-FLI1 [7]. EWS-FLI1 transcriptional repression is mediated through direct binding with the nucleosome remodeling and deactylase (NuRD) complex. The NuRD complex consists of histone deacetylases (HDACs), lysine-specific demethylase-1 (LSD1), and other DNA binding proteins and has been shown to play a role in tumor development as well as the general repression of transcription [3, 8]. Disrupting the NuRD complex through inhibition of LSD1, HDAC1, or HDAC2 may block EWS-FLI1 from affecting the transcription of oncogeneic targets. Recent studies have demonstrated that direct targeting of LSD1 with molecular tools leads to significant attenuation of cancer cell proliferation in multiple models [9–11]. Importantly, in preclinical models of ES, reversible inhibitors of LSD1 also demonstrate some promise in halting tumor cell propagation [12]. This strongly influenced our decision to select SP2509 (previously HCI2509) for our studies in lieu of irreversible LSD1 inhibitors such as GSK2879552 or ORY-1001 which have been shown to interfere with flavin adenine dinucleotide (FAD) binding which LSD1 utilizes in histone lysine methylation [13], thereby failing to reduce cell viability in the models tested [14–16]. Additionally, these catalytic inhibitors of LSD1 have been previously tested in ES and found to be insufficient as a therapeutic strategy [17, 18]. SP2509 does not interfere with FAD binding as it interacts with the H3 pocket region of LSD1 which functions as an allosteric site, suggesting that SP2509 may act as an allosteric inhibitor [14]. SP2509 is also currently in Phase I clinical testing (NCT03600649).

HDAC inhibitors have been shown to possess direct antineoplastic activity as well as to enhance the efficacy of other anticancer agents [19]. There is also evidence that inhibition of HDAC inhibition attenuates LSD1 activity *in vivo* [20]. Despite lackluster results of HDAC therapy in sarcomas, we felt this class warranted investigation as a comparator to LSD1 inhibition due to both being present in the NuRD complex [21, 22].

Due to the rarity of Ewing sarcoma, clinical trials are difficult and time consuming to conduct, increasing the need for preclinical data to direct clinical trials. By targeting the NuRD complex along with agents known to provide clinical benefit we hoped to gain insight into whether or not particular combinations of agents were synergistic or antagonistic. We previously developed a system to efficiently evaluate combinations of interest across multiple cell line models with the goal of rapid translation into relevant clinical trials [23, 24]. Our methodology has been optimized to incorporate past lessons learned from *in vitro* experiments that did not translate well in clinical applications due to unachievable lengths of exposure or metabolic restraints [25, 26]. All experimental considerations for combination therapy were developed and conducted with the end thought being the eventual clinical trial. We sought to assess LSD1 inhibition and HDAC inhibition in combination with active chemotherapies currently utilized in ES care.

## Materials and methods

### Investigational agents

Agents used included current standard of care for ES and those of experimental interest (Table 1). Due to the instability of 4-hydroperoxy cyclophosphamide (4HC) and the reversible

**Table 1. Summary of agents tested, mechanism of action, selected pharmacokinetic data, and experimental values—top concentrations and AUC at top concentrations for each drug in each cell line.**

| Agent | Mechanism of Action / Reference | Cmax (ng/ml) | AUC (h*ng/ml) | Cell Line Top Conc (ng/ml); AUC at Top Conc (ng/ml*24hr) | | | |
|---|---|---|---|---|---|---|---|
| | | | | A673 | RD-ES | TC32 | TC-71 |
| 4HC[1] | DNA Crosslinking, DNA Damage | 6927 | 27700–33000 | 2000; 48000 | 2000; 48000 | 4000; 96000 | 4000; 96000 |
| | *McCune JS, et al.(2009) J Clin Pharm; Kahen EJ, et al.(2016) Can Chem Pharm* | | | | | | |
| Doxorubicin | DNA, topo II | 2109 | 945 | 160; 3840 | 600; 14400 | 80; 1920 | 160; 3840 |
| | *Greene RF, et al.(1983) Cancer Res; Bartlett NL, et al.(1994) J Clin Oncol* | | | | | | |
| Etoposide | Topoisomerase II, DNA Damage | 20000 | 157000 | 300; 7200 | 800; 19200 | 800; 19200 | 800; 19200 |
| | *Kaul S, et al. (1995) Anticancer Drugs* | | | | | | |
| Romidepsin | Class I/II HDAC | 377 | 2414 | 4; 96 | 4; 96 | 4; 96 | 4; 96 |
| | *Fouladi M, et al.(2006) J Clin Oncol* | | | | | | |
| SN-38[2] | Topoisomerase IB | 30 | 104 | 1; 24 | 0.75; 18 | 0.75; 18 | 0.75; 18 |
| | *Ma MK, et al.(2000) Clin Cancer Res* | | | | | | |
| SP2509† | LSD1 inhibitor | 3000 | TBD | 1000; 72000 | 1000; 72000 | 2000; 144000 | 2500; 180000 |
| | *Fiskus W., et al.(2012) J Clin Oncol* | | | | | | |
| MTIC[3] | Alkylator | 13000 | 46000 | 250; 6000 | NA | 250; 6000 | NA |
| | *Horton TM, et al.(2007) J Clin Oncol* | | | | | | |
| Vincristine | Microtubules, Anti-mitotic | 40 | 90 | 2; 48 | 0.6; 14.4 | 2; 48 | 3; 72 |
| | *Guilhaumou R, et al.(2011) Cancer Chemother Pharmacol* | | | | | | |

†Value determined in rats

[1]Active metabolite of cyclophosphamide

[2]Active metabolite of irinotecan

[3]Active metabolite of temozolomide

LSD1 inhibitor SP2509, fresh drug solutions were prepared in DMSO prior to every experiment. MTIC ((5E)-5-(methylaminohydrazinylidene)imidazole-4-carboxamide, the active metabolite of temozolomide) was prepared in 100% ethanol and then mixed 1:1 with media immediately prior to application. Final ethanol concentration never exceeded 1%. Stock solutions for all other compounds were made in DMSO and stored at -20˚C. All agents were obtained directly from Selleck Chemicals (Houston, TX, USA), and Sigma-Aldrich (St. Louis, MO, USA). Structures for all agents are available in public databases.

## Cell culture

We selected four ES cell lines that are well characterized and commonly used in recent studies (Table 2) [3, 12, 27]. A673 was obtained from the ATCC (Manassas, VA). The TC32 (Children's Oncology Group (COG) Cell Line & Xenograft Repository), TC-71 (National Cancer Institute (NCI) Pediatric Preclinical Testing Program) and RD-ES cells lines were generously shared by Dr. Stephen Lessnick. A673 Cells were maintained in DMEM with 10% fetal bovine serum. TC-71, RD-ES and TC32 cells were maintained in RPMI with 15% fetal bovine serum. While protein binding can impact the activity of anti-cancer agents [28], these 15% concentrations are higher than human albumin concentrations. Media was supplemented with PenStrep prior to extended incubation times. Cells were grown at 37˚C and 5% $CO_2$. All cell lines tested

**Table 2. Summary of Ewing cell lines.**

| Cell Line | ATCC Designation | Tissue | Doubling Time (hours) | Diagnosis | Patient Info | EWS-FLI1 translocation t (11;22)(q24;q12) | FLI1-EWS Reciprocal Fusion | TP53 | KDM1A mRNA Expression Level[g] | STAG2 Status[h,i] |
|---|---|---|---|---|---|---|---|---|---|---|
| A673 | ATCC® CRL-1598™ | Muscle[a] | 25[c] | Ewing's Sarcoma[a] | 15 year old female[a] | Type 1 Fusion[d] | Detectable[d] | Non-functional (p.A119 frameshift)[c,f] | 8.49 ± 1.29 | Wildtype |
| RD-ES | ATCC® HTB-166™ | Bone[a] | 60[k] | Ewing's Sarcoma[a] | 19 year old male[a] | Type 2 Fusion[e] | Unknown | Mutant (p. R273C)[f] | 5.66 ± 0.75 | No Expression |
| TC32 | N/A | Bone[b] | 24[c] | PNET[b] | 17 year old female[b] | Type 1 Fusion[e] | Unknown | Functional (Wildtype)[c] | 2.32 ± 0.23 | I636sf |
| TC-71 | N/A | Ileum[c] | 21[c] | Ewing's Sarcoma[b] | 22 year old male[b] | Type 1 Fusion[d,e] | Undetectable[d] | Non-functional (p.R213 nonsense)[c,f] | 3.14 ± 0.50 | Wildtype |

[a]ATCC

[b]Children's Oncology Group (COG) Cell Culture and Xenograft Repository

[c]May, W.A., et al. Characterization and Drug Resistance Patterns of Ewing's Sarcoma Family Tumor Cell Lines. *PLoS One*. 2013; 8(12): e80060.

[d]Elzi, D.J., et al. The role of FLI-1-EWS, a fusion gene reciprocal to EWS-FLI-1, in Ewing sarcoma. *Genes Cancer*. 2015 Nov; 6(11–12): 452–461.

[e]Huang, H.J., et al. R1507, an Anti-Insulin-Like Growth Factor-1 Receptor (IGF-1R) Antibody, and EWS/FLI-1 siRNA in Ewing's Sarcoma: Convergence at the IGF/IGFR/Akt Axis. *PLoS One*. 2011; 6(10): e26060.

[f]Tirode, F., Sirdez, D., et al. Genomic landscape of Ewing sarcoma defines an aggressive subtype with co-association of *STAG2* and *TP53* mutations. *Cancer Discov*. 2014 Nov; 4(11): 1342–1353.

[g]Pishas, P.I., et al. Therapeutic targeting of KDM1A/LSD1 in Ewing Sarcoma with SP-2509 Engages the Endoplasmic Reticulum Stress Response. *Molecular Cancer Therapeutics*. 2018 Sep; 17(9): 1902–1916.

[h]Crompton, B.D., et al [5]

[i]Brohl, A.S., et al [4]

[k]Hyper Cell Line Database

free of mycoplasma using MycoAlert tests (Lonza Rockland, Rockland, ME). Cell line authentication was confirmed using StemElite ID system (Promega, Madison, WI) and comparing results to the ATCC, COG, and NCI short tandem repeats (STR) profiles.

## Cell viability assays

The activity of drugs alone and in combination was determined by a high-throughput cell viability assay as described previously [23, 24]. Cells ($4.5 \times 10^3$ per well) were transferred to 384-well plates and incubated for 24 hours prior to drug administration, which was empirically confirmed to be a period of log-phase growth for these conditions. At 72 hours, cell viability was assessed using CellTiter-Glo (CT-Glo) (Promega, Madison, WI, USA), which provides luminescence proportional to cellular ATP levels. This reflects the reduction in cellular metabolism (an indication of cell viability) due to the drug treatment. Data were transferred to custom Microsoft Excel templates and the percent viability/fraction affected was calculated by normalization to untreated control conditions. Specifically, FA is calculated as FA = 1 −(CT-glo signal with drug treatment)/(CT-glo signal without drug treatment). Absolute IC50 values were determined using sigmoidal equilibrium model regression and fitted using XLfit version 5.5 (ID Business Solutions, Guildford, Surrey, England). The IC50 values obtained from single-drug viability assays were used to design subsequent drug combination experiments.

## Single-agent screening

Single agent dose response was characterized for a panel of 7 therapeutic candidates across 4 ES cell lines (A673, RD-ES, TC32 and TC-71) (Table 2). Human pharmacokinetic data was collected using pediatric and combination studies when available from previously reported Phase I trials. The majority of drugs selected (excluding only romidepsin and SP2509) have half-lifes greater than 8 hours or have a continuous dosing schedule clinically. The half-life of SP2509 is unknown in humans. We chose to uniformly expose the cells for 72 hours to all the drugs. Experiments were performed in triplicate with at least 4 technical replicates per biological replicate.

## Two-drug combination screening

A 5x5 checker-board matrix was used to assess all two-drug combinations at five clinically achievable concentrations. Each combination was evaluated at multiple drug ratios to identify synergy. Analysis of additive and synergistic effects was done by measuring cell viability with the CellTiter-Glo assay with results analyzed using the Combination Index (CI) method of Chou-Talalay [29]. CI, a measure of drug synergy, is derived by taking the dose-effect curve for each drug using the median effect principle and then comparing it to the effect achieved with the 2-drug combination. 2-drug combination activity that is merely additive is represented by a CI value of 1.0. As CI decreases below 1.0, drug synergism increases. Antagonistic drug effects are indicated by a CI greater than 1.0. A minimum of two biological replicates per cell line were performed with at least 4 technical replicates per condition in each experiment.

## Order of addition assays

A673 and TC32 cells ($2.25 \times 10^3$ and $4.5 \times 10^3$ per well, respectively) were transferred to 384-well plates and incubated for 24 hours prior to initial drug administration. Cells were then treated with one, two or three drugs concurrently at the chosen top concentration, one half, and one quarter of the top concentration, and incubated for an additional 24 hours. Experimental plates were then read on Day 2 with RealTime-Glo (RT-Glo) (Promega, Madison, WI, USA) (proprietary technology) before being treated with an additional drug. Plates were then read a second time 72 hours later on Day 5, and final cell viability results were obtained. Experiments were performed in duplicate plus an additional run with higher cell numbers with 12 technical replicates per condition in each experiment.

## Statistical methods

The drug response data was modeled using a sigmoidal equilibrium regression curve using the software package XLfit version 5.5. Differences between drug response measurements (i.e., FA values) between drug combinations was completed using linear mixed effects model with a random intercept for cell lines to account for multiple measurements taken off the same cell line with a fixed effect adjustment for concentration level of the drug using R package *lme4*. For comparisons between the 21 drug combinations a Bonferroni adjustment for multiple testing was used.

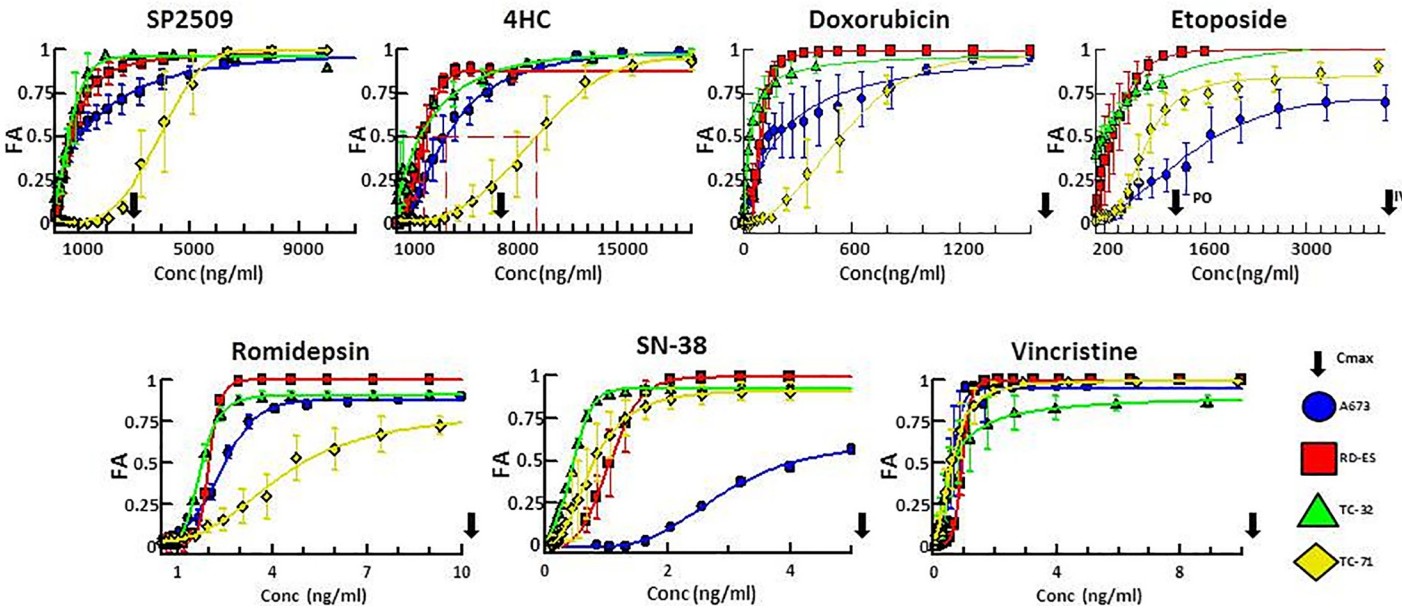

**Fig 1. Single agent dose response plots for 7 tested agents against 4 ES cell lines demonstrate efficacy at clinically achievable concentrations.** Dose response plots of each drug tested across 4 cell lines plotted as fraction affected (FA) versus concentration (ng/ml). Data was recovered 72 hours after drug treatment. Plotted points represent the mean FA values while error bars represent ±SEM (standard error of the mean) between technical replicates. Arrows beyond the x-axis indicate a Cmax in excess of the values displayed.

## Results

### NuRD complex directed therapeutics demonstrate considerable *in vitro* activity at clinically achievable levels

We characterized the single-agent activity of a panel of 7 therapeutic candidates (Table 1) using 4 ES cell lines (A673, RD-ES, TC32 and TC-71) (Table 2). The active metabolite of the alkylating prodrug cyclophosphamide, 4HC; the DNA intercalator and topoisomerase II inhibitor, doxorubicin; the topoisomerase II inhibitor, etoposide; the active metabolite of the topoisomerase I inhibitor irinotecan, SN-38; and the Beta-tubulin inhibitor, vincristine were chosen due to their role in first and second line treatment of ES. The additional 2 agents, the HDAC1 and HDAC2 inhibitor romidepsin and the reversible LSD1 inhibitor SP2509, were selected based on their respective targets in the NuRD complex [3]. To evaluate the potency of these agents, full dose-response curves were obtained for the drug panel in each cell line. As anticipated, cell lines showed sensitivity to agents used clinically in the management of ES. In addition, experimental candidates romidepsin and SP2509 demonstrated potency in the single agent context. The half maximal inhibitory concentrations (IC50s), simulated maximal plasma concentrations (Cmax), and area under the curves (AUC) were within clinically achievable levels for all drugs selected in nearly all contexts (Fig 1, Table 1, Table 2). Some variability between cell lines was observed with A673 and TC-71 demonstrating a trend towards tolerance relative to RD-ES and TC32; however, cell line drug sensitivities were mostly uniform across the cell lines and were uniform for vincristine. A673 demonstrated a relative tolerance to topoisomerase inhibitors while TC-71 was more tolerant to alkylators and epigenetic agents (S1 Table).

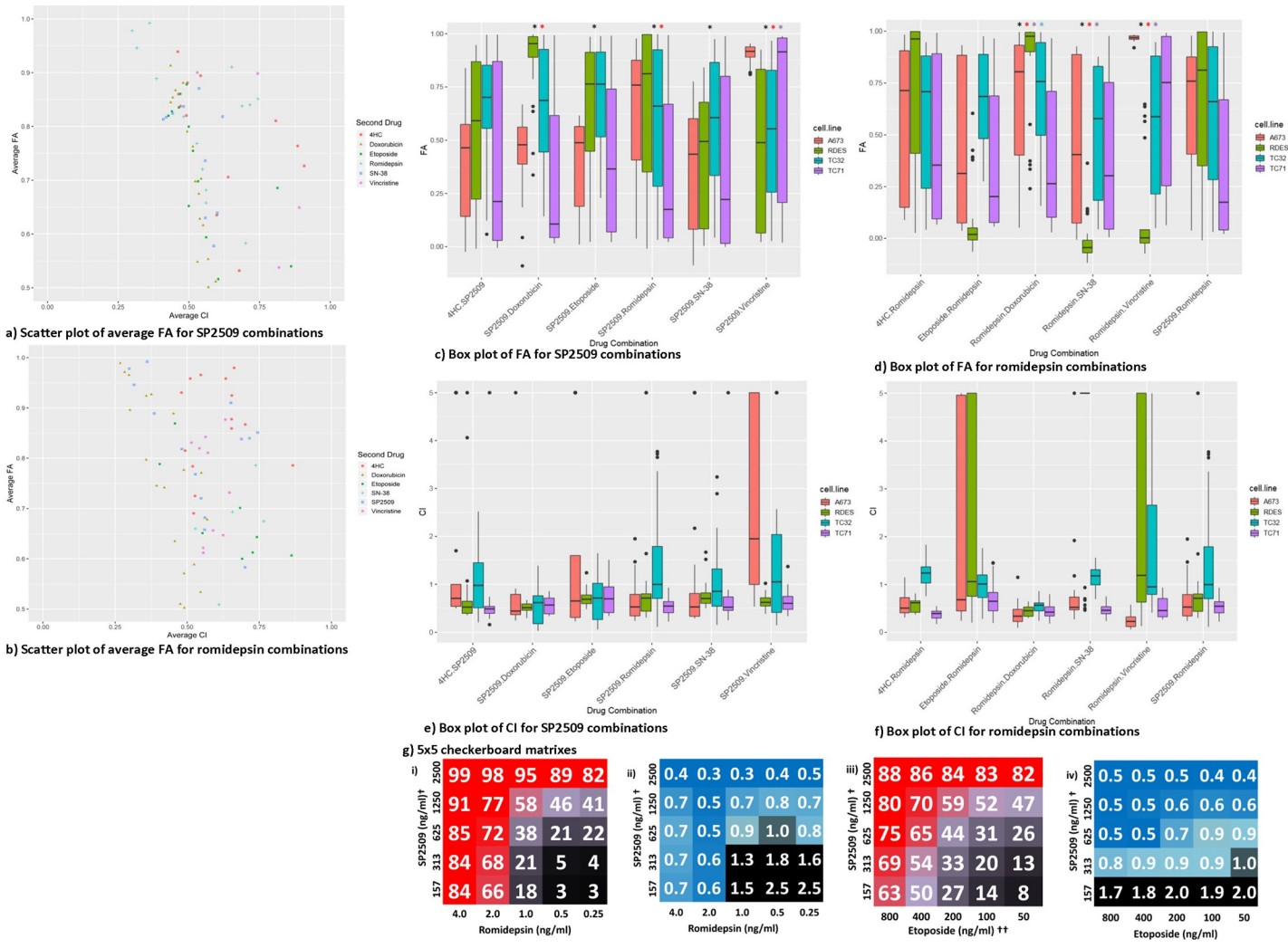

**Fig 2. Overview of combination efficacy for SP2509 and romidepsin combinations.** Scatter plots displaying the average fraction of cells affected (FA) for drugs paired with **a)** SP2509 and **b)** romidepsin. Box plot for **c)** SP2509 exhibiting the FA of each cell line for all drug combinations. A black asterisk indicates a significantly higher FA value than 4HC+SP2509. P-values as follows: SP2509+doxorubicin = 1.87x10$^{-25}$, SP2509+etoposide = 2.54x10$^{-16}$, SP2509+romidepsin = 6.72x10$^{-15}$, SP2509+SN-38 = 2.81x10$^{-08}$, SP2509+vincristine = 1.02x10$^{-18}$. A red asterisk indicates a significantly higher FA value than SP2509+etoposide. P-values as follows: SP2509+doxorubicin = 1.66x10$^{-09}$, SP2509+romidepsin = 3.43x10$^{-06}$, SP2509+vincristine = 1.28x10$^{-11}$. A purple asterisk indicates a significantly higher FA than SP2509+doxorubicin. P-value as follows: SP2509+Vincristine = 3.19x10$^{-05}$. Statistics based on combined value from all cell lines and all concentrations. Box plot for **d)** romidepsin exhibiting the FA of each cell line for all drug combinations. A black asterisk indicates significantly higher FA than romidepsin+etoposide. P-values as follows: romidepsin+doxorubicin = 8.56x10$^{-07}$, romidepsin+SN-38 = 6.97x10$^{-07}$, romidepsin+vincristine = 7.23x10$^{-19}$. A red asterisk indicates significantly higher FA than romidepsin+SP2509. P-value as follows: romidepsin+doxorubicin = 0.00011, romidepsin+vincristine = 2.26x10$^{-12}$. A purple asterisk indicates significantly higher FA value than romidepsin+4HC. P-value as follows: romidepsin+vincristine = 8.11x10$^{-11}$. Statistics based on combined value from all cell lines and all concentrations. Box plots for **e)** SP2509 and **f)** romidepsin exhibiting the CI for each cell line for all drug combinations. **g)** 5x5 Checkerboard matrixes assessing combination activity. **i)** FA and **ii)** CI values for SP2509 and romidepsin assessed at 25 different concentrations. †Alternate SP2509 concentrations used (1000, 500, 250, 125, 62.5ng/ml for A673 and RD-ES cell lines; 2000, 1000, 500, 250, 125ng/ml used for TC32 cell line). Variable concentrations used due to differences in IC$_{50}$ per cell line. **iii)** FA and **iv)** CI values for SP2509 and etoposide assessed at 25 different concentrations. ††Alternate etoposide concentrations used (300, 150, 75, 37.5, 18.8ng/ml used for A673 cell line).

## The LSD1 inhibitor SP2509 and the HDAC inhibitor romidepsin are synergistic with conventional chemotherapeutic agents *in vitro*

We hypothesized that agents that interact with components of the NuRD complex would work synergistically with agents already utilized in the treatment of ES. We performed high throughput cell viability assays that assessed each of the 7 agents in combination at 25 different

concentration pairs. All drugs were tested below Cmax values and spanned the IC50 for each cell line to allow for synergy (combination index, CI) calculations. We identified multiple drug combinations that demonstrated high fraction affected (FA) values (see methods for further discussion of CI and FA). The FA and CI values of each drug combination and drug ratio in each individual cell line are summarized in S2 Table in order of top performing combinations. The average distribution of effects for each drug combination trends toward high FA and low CI values when paired with SP2509 or romidepsin. (Fig 2A and 2B, respectively). SP2509 generally showed synergy with all drugs apart from vincristine which demonstrated antagonism at most concentrations in this combination. Combinations containing romidepsin, while still trending towards high FA and low CI, were less synergistic than those with SP2509 overall, experiencing antagonistic drug interactions with vincristine as well as etoposide. In order to gain a comprehensive overview of FA across all cell lines and concentrations, statistical analysis was performed using a linear mixed effects model with a random intercept for cell lines and a Bonferroni adjustment for multiple testing (adjusting for 210 tests, a p-value of 0.0002 is considered significant; see methods for further discussion of statistical analysis). Our analysis showed that of the six possible combinations with SP2509, all the combinations produced a more significant FA than SP2509+4HC (Fig 2C). Combinations SP2509+romidepson, SP2509 +doxorubicin, and SP2509+vincristine all produced FAs more statistically significant than SP2509+etoposide. And the FA of the SP2509+vincristine was more statistically significant than the combination of SP2509 with 4HC, etoposide, and doxorubicin. When the same analysis was performed with romidepsin combinations, we found romidepsin+doxorubicin, romidepsin+SN-38, and romidepsin+vincristine to have the most statistically significant FAs (Fig 2D). Romidepsin+vincristine had the most statistically significant FAs, more significant than when paired with etoposide, SP2509, and 4HC. Interestingly, nearly all the combinations produced higher FA values than those produced with romidepsin+SP2509.

When viewed comparatively between cell lines, sensitivity differences to combinations containing SP2509 or romidepsin were observed (Fig 2C, 2D, 2E and 2F, S2 Table). A673 was typically more resistant to drug pairs with SP2509 (Fig 2C). Except for SP2509+vincristine, FA values trended lower than the other cell lines, though this is likely due more to A673's sensitivity to vincristine. This sensitivity is likely to have skewed the statistical analysis for this combination, making the FA appear more significant for this combination than its actual performance across all the cell lines. RD-ES and TC32 were similar in their drug tolerances, mostly demonstrating high median and maximum FA values. TC-71 had the largest distributions of effects across all the combinations tested, though kill rates near 100% (FA near 1.0) were still reached for all combinations with SP2509. The distribution of CI values for SP2509 combinations was similar in RD-ES and TC-71 with median values between 0.5 and 0.8 (Fig 2E). SP2509 synergism in TC32 was more greatly distributed showing antagonism at some concentrations and synergism at others. FA values for combinations containing romidepsin were more mixed (Fig 2D). The most notable differences compared to SP2509 combinations include the relative resistance of RD-ES and the notable potency of romidepsin+vincristine in A673. Synergism was consistently observed in combinations containing romidepsin with TC32 displaying higher values at more concentrations than in other lines (Fig 2F). The resistance of RD-ES to these combinations is associated with highly distributed CI values. The combination of SP2509 with romidepsin itself was rather effective, producing median FA values near and above 0.7 across all cell lines except TC-71, with maximum FA values of 1.0 reached in all 4 ES models (Fig 2A, 2B, 2C, 2D, 2G and 2I). Synergy between the drugs was also consistently seen (average CI = 0.4), with TC32 having a bit of a larger distribution across all concentrations tested compared to the other lines (Fig 2B, 2E, 2F and 2Gi). Notably, even at lower concentrations of both drugs, high FA and low CI values were typically still observed.

Topoisomerase inhibitors showed particularly strong activity and synergy when paired with SP2509. SP2509 and any of doxorubicin, etoposide or SN-38 demonstrated an average FA of at least 0.87, with FA as high as 0.95 observed in two cell lines and average CIs of 0.5 or lower (Fig 2A, 2C, 2E and 2Giii,iv, S2 Table). We were particularly interested in the translational potential of combining SP2509 with etoposide since oral etoposide is often used in the relapsed setting, continuously dosed, and well tolerated in patients. At lower concentrations of etoposide, combined with the top concentration of SP2509, FA was 0.88 (Fig 2Giii) and minimum CI values of 0.4 (Fig 2Giv) were obtained. Synergy was achieved at all except the lowest concentrations of SP2509.

### SP2509 enhances efficacy of VIT (vincristine, irinotecan, and temozolomide)

In order to further investigate the translational potential of SP2509, we administered it in combination with the conventional treatment utilized in relapsed ES, VIT: Vincristine, Irinotecan (SN-38, the active metabolite), and temozolomide (MTIC, the active metabolite)[30, 31]. We chose two cell lines, A673 and TC32, which we felt would be representative of our cell lines and had diversity amongst the other genes that may play a role in eventual subtyping of Ewing sarcoma, *TP53* and *STAG2*. TC32 has a functional p53 gene while A673 has a frameshift mutation rendering p53 nonfunctional. Conversely, A673 is wild type for *STAG2* whereas TC32 has a frameshift mutation in *STAG2* (Table 1) [4, 27, 32]. We assayed for potential differences in potency with this 4-drug combination by testing multiple orders of addition of the agents as well as one half and one quarter concentrations of what we estimated would be an optimal top concentration (Fig 3A and 3B, S1 Fig). Utilizing RT-Glo, we were able to obtain multiple time-points of the same experimental plate, which allowed us to observe drug kinetics over time. We found that the effects on viability from SN-38 were more delayed than any of the other agents. This was most noticeable in A673 where there was a 50-fold shift in the IC50 between 24-hours and 96-hours (40ng/ml vs 0.77ng/ml, respectively) (Table 3). The practical result of this is that combination FAs were typically higher when SN-38 was added first rather than second (Fig 3A and 3B; S1 Fig). TC32 appears to be more sensitive overall with the exception of vincristine, though this may simply be due to the differences in doubling times. Indeed, because of the speed of A673 proliferation, cell numbers needed to be halved to make it compatible with this protocol due to considerable variability between experiments (S1C Fig). The particular sensitivity of A673 to vincristine in this context is worth further study. Importantly, at concentrations well below Cmax (Table 3, S1A Fig), combinations of VIT+SP2509 were able to reduce cell viability by over 90% at the highest-tested concentrations and by over 80% at half the topmost concentrations. It should be noted that due to the instability of MTIC, the actual concentrations in media are likely below the calculated estimate. In A673, there is no clear preference for any particular order of addition with these 4 agents, though the largest increase between Day 2 and Day 5 was seen when SP2509 was added last. In TC32, there is a tendency towards higher potency when SP2509 is added last, but this is within the bounds of experimental variability. Overall, addition of SP2509 increased the FA but only slightly, most likely due to the already efficacious treatment of VIT alone. However, this increase can be noted at lower concentrations as well as when SP2509 is substituted for one of the other drugs in combination (S1 Fig).

## Discussion

In this study, our aim was to evaluate the inclusion of romidepsin and SP2509, both NuRD complex-directed therapeutics, with agents currently utilized in the treatment of ES. Our results confirm that ES cell lines are sensitive to multiple chemotherapeutic agents commonly

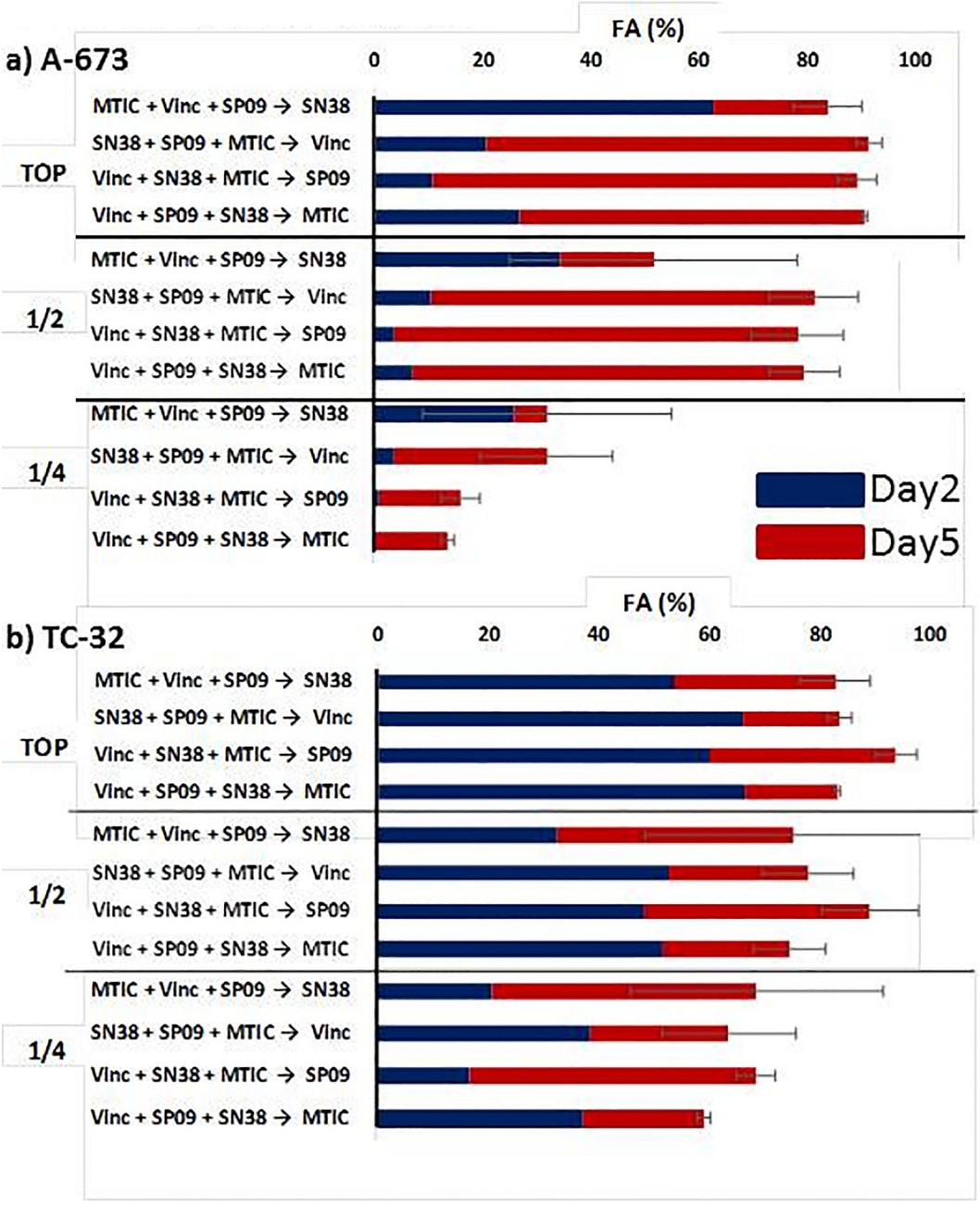

**Fig 3. Vincristine, irinotecan, temozolomide, and SP2509 order of addition.** Results of order of addition experiments for **a)** A673 and **b)** TC32, respectively. One to three drugs were given concurrently 24 hours after plating and read on Day 2. The fourth drug was added after the 48 hour read and the same experimental plate was read on Day 5. TOP represents the highest concentration used as indicated in Table 3B; ½ is half the highest concentration, ¼ is a quarter of the highest concentration. Error bars represent standard error of the mean. A673 n = 2, TC32 n = 4. See S1 Fig for all order of addition results.

used in ES treatment and the addition of NuRD complex-directed drugs show promising results as potential additions to future chemotherapeutic regimens. A summary of the top performing agents is included in S3 Table. When considering promising agents for ES, the practicalities of the chemotherapeutic regimen should be considered early in order to best inform eventual trial approaches which will incorporate a new agent. Standard initial therapy for ES

**Table 3. Single agent dose response IC$_{50}$s at 24-hours and 96-hours after treatment, ± standard error of the mean.**

| Tx | A673 | | TC32 | | Top Concentration | Cmax (ng/ml) |
|---|---|---|---|---|---|---|
| | Day 2 | Day 5 | Day 2 | Day 5 | | |
| MTIC | 18889 ± 3240 | 18915 ± 3672 | 15547 ± 7867 | 9447 ± 4045 | 250 | 276 |
| SN38 | 40 ± 33 | 0.77 ± 0 | 4.0 ± 0.5 | 2.6 ± 0.2 | 4 | 30 |
| SP2509 | 4222 ± 863 | 1171 ± 212 | 2375 ± 221 | 1238 ± 89 | 1000 | 3000 |
| Vincristine | 0.73 ± 0.25 | 0.79 ± 0.03 | 6.2 ± 1.4 | 2.0 ± 0.3 | 1 | 40 |

includes a clinical trial when available and in the United States typically is based on every 2 week therapy with vincristine, doxorubicin and cyclophosphamide alternating with ifosfamide and etoposide [33]. Relapse studies typically incorporate a camptothecin (either topotecan or irinotecan) with an alkylating agent (typically cyclophosphamide or temozolomide respectively) or oral etoposide when a trial is not available [34].

As of this writing, optimal schedule, toxicities, adverse effects and maximal tolerated doses with resultant PK are not yet available for reversible LSD1 inhibitors in humans. Trials are open with SP2577, a similar compound, in the Ewing sarcoma population with continuous dosing schedules, NCT03600649. Importantly, when SP2509 is combined with current second-line conventional chemotherapy regimen vincristine, irinotecan, temozolomide (VIT) *in vitro*, cell viability is considerably decreased even at concentrations well below Cmax. The concentrations used in these experiments are well below the observed maximal plasma concentrations observed in pharmacokinetic assays and thus it is possible that sufficient efficacy could be achieved at lower than maximally tolerated doses, which may minimize side effects and maximize additional agents being combined in the future. 4HC, etoposide, doxorubicin, and SN-38 all exhibited robust activity alone and in combination with SP2509, which was confirmed when we performed a correlation analysis on the FA of all combinations tested (Fig 4). If indeed SP2509 is acting as an LSD1 inhibitor, LSD1 is required for heterochromatin formation [35, 36]. When inhibited, DNA would be structurally less dense and more open to agents that damage DNA such as topoisomerases, which correlates well with our findings.

Of note is the antagonism exhibited when SP2509 and vincristine are paired in the A673 cell line. A possible explanation is the unique sensitivity of A673 to vincristine, disallowing synergy. Inhibition of the EWS-FLI1 fusion protein has been shown to decrease EWS-FLI1--mediated generation of microtubule-associated proteins leaving cells more susceptible to microtubule depolymerization by vincristine [37]. Real time quantitative PCR for the EWS/ETS gene product as well as the native FLI1 showed the lowest expression of EWS/ETS in A673, followed by TC32 and then TC-71 (RD-ES was not tested) [38]. However, it has also been shown that levels of the EWS-FLI1 transcript and protein are heterogeneous from one cell to another and can range from low to high expression levels and fluctuate along time, producing different phenotypes of cell proliferation, migration, invasion and metastases [39]. Clearly more investigation is required before a definitive assessment of these mechanisms can be ascertained. Another possibility is that the cell lines have relative resistance to vincristine from prior exposure. Indeed, TC-71 and TC32 were derived post chemo, RD-ES was derived from a primary osseous of the humerus, and it is unknown whether A673 was derived pre- or post-chemotherapy. A673 was derived in 1973 and it is known that lower doses of chemotherapy were administered then, but vincristine most likely would have been used [38]. This does not seem to be generalizable to all microtubule inhibitors as when SP2509 was administered with docetaxel to prostate cells, synergy is noted at sub-IC$_{50}$ doses [40].

Another possible translational route would be combining an epigenetic agent with oral etoposide. This route has the advantages of long exposures of both agents at effective

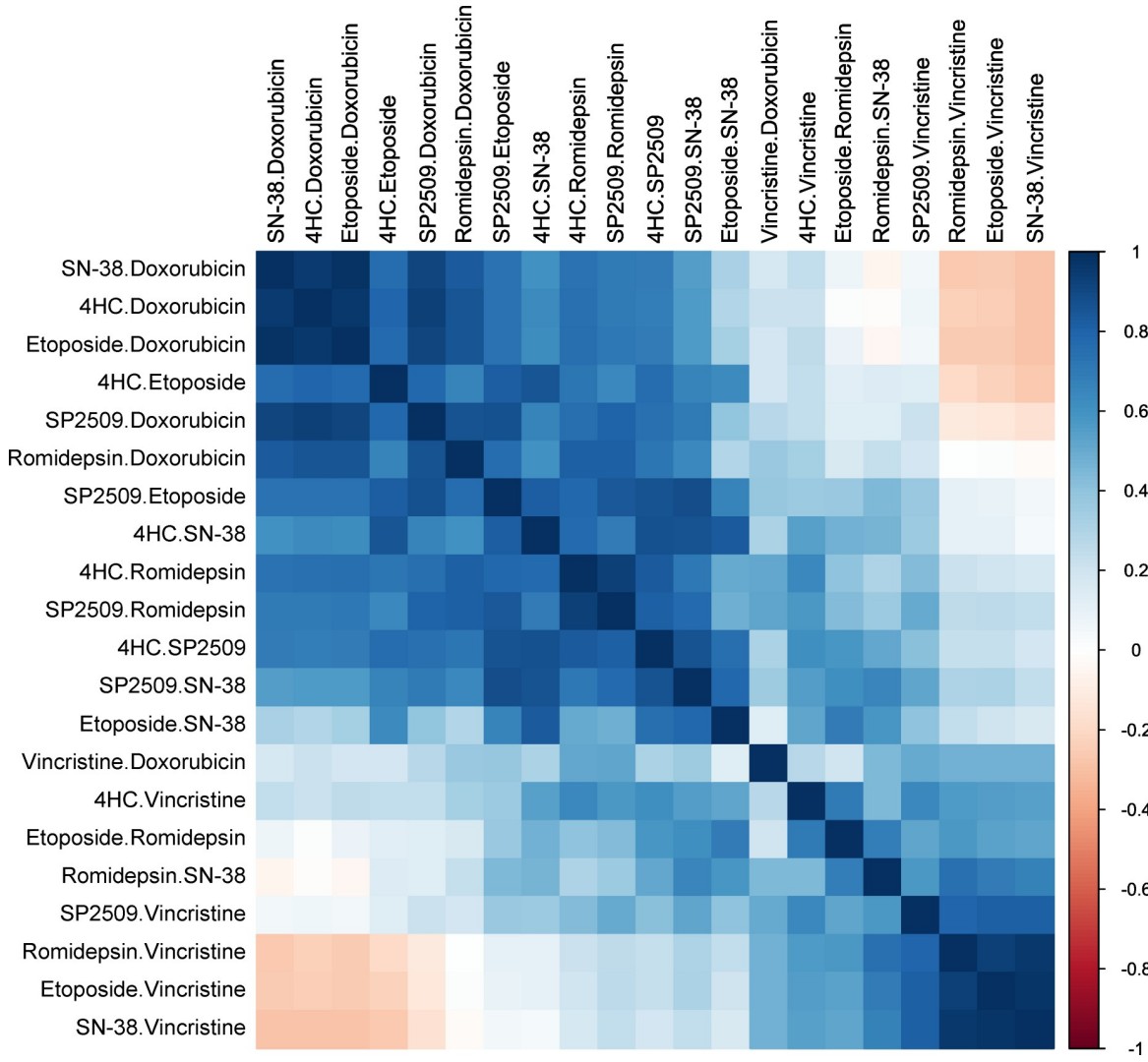

**Fig 4. Correlation analysis of FA for all combinations tested.** Spearman correlation between FA values from four cell lines treated with the various drug combinations. Blue values indicate strong positive correlation while red values indicate strong negative correlation between drug FA values.

concentrations along with the fewest side effects of the chemotherapeutic backbone. An additional potential advantage of combination strategies utilizing SP2509 beyond its distinct mechanism of action is the potential feasibility of continuous dosing. We postulated a clinical dosing schedule for one week with oral etoposide paired with SP2509 and romidepsin (Fig 5). In the case of SP2509 and etoposide, multiple opportunities for synergism exist at the concentrations used and times administered. Recent combination data with vincristine and translocation targeting agents such as TK216 are also being reported [37]. The findings here of synergy and activity at low levels are especially interesting due to their clinical applications.

We demonstrated that the HDAC inhibitor romidepsin would also be a promising addition to standard chemotherapy agents utilized in relapsed or refractory Ewing sarcoma. Even though multiple opportunities for synergism do not exist as in the case of etoposide and SP2509, romidepsin has been shown here to be effective even at lower concentrations and could prove synergistic beyond the levels we tested. Epigenetic modification is particularly of

**SP2509 PO (1) / Etoposide PO (2)**

| | ng/ml | D1 | | D2 | | D3 | | D4 | | D5 | | D6 | | D7 | |
|---|---|---|---|---|---|---|---|---|---|---|---|---|---|---|---|
| SP2509 PO¹ | 2442 | | | | | | | | | | | | | | |
| Rat PK only available | 1221 | 70 | | 70 | | 70 | | 70 | | 70 | | 70 | | 70 | |
| Theoretical | 610.5 | 44 | | 44 | | 44 | | 44 | | 44 | | 44 | | 44 | |
| Half life 4hrs | 305.3 | 20 | 20 | 20 | 20 | 20 | 20 | 20 | 20 | 20 | 20 | 20 | 20 | 20 | 20 |
| | 152.6 | 8 | 8 | 8 | 8 | 8 | 8 | 8 | 8 | 8 | 8 | 8 | 8 | 8 | 8 |

| | ng/ml | D1 | | | | D2 | | | | D3 | | | | D4 | | | | D5 | | | | D6 | | | | D7 | | | |
|---|---|---|---|---|---|---|---|---|---|---|---|---|---|---|---|---|---|---|---|---|---|---|---|---|---|---|---|---|---|
| Etoposide PO² | 600 | | | | | | | | | | | | | | | | | | | | | | | | | | | | |
| 50 mg/m2 | 300 | 70 | | | | 70 | | | | 70 | | | | 70 | | | | 70 | | | | 70 | | | | 70 | | | |
| days 1-21 of 28 | 150 | 44 | 33 | | | 44 | 33 | | | 44 | 33 | | | 44 | 33 | | | 44 | 33 | | | 44 | 33 | | | 44 | 33 | | |
| Half life 6.5hrs | 75 | 20 | 20 | x | | 20 | 20 | x | | 20 | 20 | x | | 20 | 20 | x | | 20 | 20 | x | | 20 | 20 | x | | 20 | 20 | x | |
| | 37.5 | 8 | 8 | x | x | 8 | 8 | x | x | 8 | 8 | x | x | 8 | 8 | x | x | 8 | 8 | x | x | 8 | 8 | x | x | 8 | 8 | x | x |

**Romidepsin (3) / Etoposide PO**

| | ng/ml | | | | | |
|---|---|---|---|---|---|---|
| Romidepsin³ | 1000 | | | | | |
| 14 mg/m2 | 500 | | | | | |
| IV days 1,8,15 | 250 | x | | | | |
| Half life 3hrs | 125 | x | | | | |
| | 62.5 | x | x | | | |
| | 31.3 | x | x | | | |
| | 15.7 | x | x | x | | |
| | 7.6 | x | x | x | | |
| | 3.8 | 70 | 61 | 60 | 60 | |
| | 1.9 | 50 | 50 | 46 | 40 | |
| | 1 | 65 | 24 | 22 | 18 | 65 |
| | 0.5 | 50 | 16 | 13 | 13 | 50 |

| | ng/ml | D1 | | | | D2 | | | | D3 | | | | D4 | | | | D5 | | | | D6 | | | | D7 | | | |
|---|---|---|---|---|---|---|---|---|---|---|---|---|---|---|---|---|---|---|---|---|---|---|---|---|---|---|---|---|---|
| Etoposide PO | 600 | | | | | | | | | | | | | | | | | | | | | | | | | | | | |
| 50 mg/m2 | 300 | 70 | | | | 21 | | | | x | | | | x | | | | x | | | | x | | | | x | | | |
| days 1-21 of 28 | 150 | 61 | 61 | | | 16 | x | | | x | x | | | x | x | | | x | x | | | x | x | | | x | x | | |
| Half life 6.5hrs | 75 | 60 | 60 | 60 | | 13 | x | x | | x | x | x | | x | x | x | | x | x | x | | x | x | x | | x | x | x | |
| | 37.5 | 60 | 60 | 60 | 60 | 12 | x | x | x | x | x | x | x | x | x | x | x | x | x | x | x | x | x | x | x | x | x | x | x |

CI<0.6
CI=0.6-0.9
CI>0.9

1. Fiskus W., et al. (2012) J Clin Oncol
2. VePesid (etoposide) for injection and capsules [package insert] Bristol-Myers Squibb Co., Princeton, NJ; 2004. 3. Istodax® (romidepsin) for injection [package insert] Gloucester Pharmaceuticals, Inc., Cambridge, MA; 2009.

**Fig 5. Clinical schedules and corresponding PK predicted activities for combinations of interest.** Dosing schedules for a 7-day treatment of oral etoposide paired with SP2509 and romidepsin. Plasma concentrations and estimated half-life for single-drug administration are derived from sources listed. Values in red indicate plasma concentrations comparable to experimental concentrations used. Values in boxes represent expected FA at given concentrations. Color represents amount of synergy expected. Gray boxes represent untested concentration combinations.

interest in a malignancy like ES which is characterized by a translocation altering transcriptional control. HDACs are involved in a broad number of biological pathways, and interruption of their function can result in a plethora of transcriptional and functional consequences [41]. Sankar et. al. demonstrated that transcriptional repression of EWS-FLI1 is mediated through direct binding of the NuRD complex and that NuRD-associated HDAC and LSD1 functions are vital to this repression [3].

Of additional interest is the fact that we also saw synergistic activity when SP2509 was paired with romidepsin. These findings are consistent with other studies that investigated the co-administration of LSD1 and HDAC inhibitors in other cancer models as well as recent developments with a dual LSD-1/HDAC hybrid inhibitor [19, 42–44]. Although the most significant FA values we obtained with romidepsin were with drugs that utilized a different

mechanism of action, our findings of synergism with SP2509 suggest that the addition of both agents into the current ES treatment could prove beneficial, but further investigation is required.

The methods presented here demonstrate a comprehensive, reproducible, and high-throughput method for exploring antitumor effects of combinations of therapies at clinically achievable concentrations. In particular we discovered that combinations of SP2509 with currently utilized conventional chemotherapies demonstrate largely synergistic activity against ES cell lines. Several combinations seem to have translational promise. Combining SP2509 with oral etoposide would maximize the potential time that both agents can be maintained at therapeutic levels. Combining with VIT would be of interest as well and consideration for timing of SP2509 should be further considered. Additional explorations of these combinations through available murine models are recommended as well as extended assays to see the effects of long-term drug treatment.

## Supporting information

**S1 Fig. All order of addition results including two-drug, three-drug, and four-drug combinations as discussed in Fig 3. a)** A673 (2250 cells), **b)** TC32 (4500 cells), **c)** A673 (4500 cells).
(TIF)

**S1 Table.** A) Average $IC_{50}$ ± standard error. B) $IC_{50}$ ± standard error as a % of Cmax. C) $IC_{50}$ ± standard error with conditional formatting indicating % Cmax.
(TIFF)

**S2 Table. Full table of two-drug combinations.**
(DOCX)

**S3 Table. Top combination treatment regimens based on a combination of FA and CI.**
(TIF)

## Acknowledgments

We thank Dr. Kathleen Pishas for critical review of the manuscript.

## Author Contributions

**Conceptualization:** Darcy Welch, Elliot Kahen, Andrew S. Brohl, Christopher L. Cubitt, Damon R. Reed.

**Data curation:** Darcy Welch, Elliot Kahen, Brooke Fridley, Damon R. Reed.

**Formal analysis:** Darcy Welch, Elliot Kahen, Brooke Fridley.

**Funding acquisition:** Damon R. Reed.

**Investigation:** Darcy Welch, Elliot Kahen.

**Methodology:** Darcy Welch, Elliot Kahen, Brooke Fridley, Andrew S. Brohl, Christopher L. Cubitt, Damon R. Reed.

**Project administration:** Damon R. Reed.

**Resources:** Damon R. Reed.

**Supervision:** Damon R. Reed.

**Validation:** Elliot Kahen.

**Visualization:** Darcy Welch, Elliot Kahen, Brooke Fridley.

**Writing – original draft:** Darcy Welch.

**Writing – review & editing:** Darcy Welch, Elliot Kahen, Brooke Fridley, Andrew S. Brohl, Christopher L. Cubitt, Damon R. Reed.

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
