## [Decision Letter · Decision Letter 0]

29 Jul 2019

PONE-D-19-18074

Small Molecule Inhibition of Lysine-Specific Demethylase 1 (LSD1) and Histone Deactylase (HDAC) Alone and in Combination in Ewing Sarcoma Cell Lines

PLOS ONE

Dear Dr Reed,

Thank you for submitting your manuscript to PLOS ONE. After careful consideration, we feel that it has merit but does not fully meet PLOS ONE’s publication criteria as it currently stands. Therefore, we invite you to submit a revised version of the manuscript that addresses the points raised during the review process.

Although both the reviewers indicated an overall enthusiasm for the studies, it was however felt that further discussion and justifications/rationale for choosing certain compounds and their combination studies would be necessary. 

We would appreciate receiving your revised manuscript by Sep 12 2019 11:59PM. To enhance the reproducibility of your results, we recommend that if applicable you deposit your laboratory protocols in protocols.io, where a protocol can be assigned its own identifier (DOI) such that it can be cited independently in the future. For instructions see: http://journals.plos.org/plosone/s/submission-guidelines#loc-laboratory-protocols

We look forward to receiving your revised manuscript.

Kind regards,

Arun Rishi, Ph.D.

Academic Editor

PLOS ONE

2. Please provide additional information about each of the cell lines used in this work, including history, culture conditions and any quality control testing procedures (authentication, characterisation, and mycoplasma testing). For more information, please see http://journals.plos.org/plosone/s/submission-guidelines#loc-cell-lines.

3. Our staff editors have determined that your manuscript is likely within the scope of our Targeted Anticancer Therapies and Precision Medicine Call for Papers. This editorial initiative is headed by a team of Guest Editors for PLOS ONE: Andrew Cherniack, Anette Duensing, Steven Gray, Sunil Krishnan, Chandan Kumar-Sinha and Gayle Woloschak. The Collection will encompass a diverse range of research articles about the identification and classification of driver genes and somatic alterations, target and drug discovery, mechanisms of drug resistance, and early detection and screening.  Additional information can be found on our announcement page: https://collections.plos.org/s/targeted-anticancer-therapies.

If you would like your manuscript to be considered for this collection, please let us know in your cover letter and we will ensure that your paper is treated as if you were responding to this call. If you would prefer to remove your manuscript from collection consideration, please specify this in the cover letter.

4. Thank you for stating the following in the Financial Disclosure section: "This study was generously supported by the National Pediatric Cancer Foundation (www.nationalpcf.org). This work has been supported in part by the Translational Research Core at the H. Lee Moffitt Cancer Center & Research Institute, a NCI designated Comprehensive Cancer Center (P30-CA076292)."

We note that one or more of the authors are employed by a commercial company: 'Sunshine Lab LLC'.

Reviewers' comments:

Reviewer's Responses to Questions

**Comments to the Author**

1. Is the manuscript technically sound, and do the data support the conclusions?

Reviewer #1: Yes

Reviewer #2: Yes

2. Has the statistical analysis been performed appropriately and rigorously? 

Reviewer #1: Yes

Reviewer #2: Yes

3. Have the authors made all data underlying the findings in their manuscript fully available?

Reviewer #1: Yes

Reviewer #2: Yes

4. Is the manuscript presented in an intelligible fashion and written in standard English?

Reviewer #1: Yes

Reviewer #2: Yes

5. Review Comments to the Author

Reviewer #1: The purpose of this study by Welch et al., was to define clinical agents that synergize with the LSD1 inhibitor SP2509 in Ewing sarcoma, an aggressive childhood cancer which has seen limited improvement in overall survival since the introduction of chemotherapy >50 years ago. The authors tested 7 clinically utilized chemotherapeutic agents in four different Ewing sarcoma cell lines with varying KDM1A expression levels and TP53/STAG2 mutational status. Order of addition of second line conventional combination therapy agents were also tested with the addition of SP2509.

The authors should be commended on several factors including their reproducible and high throughput method for exploring antitumor effects of combinations of therapies at clinically achievable concentrations. All concentrations used in experiments were well below observed maximal plasma concentrations. In addition, it is refreshing to see that the authors used active metabolites of cyclophosphamide, irinotecan, and temozolomide which is frequently overlooked. As all experimental considerations for combination therapy were developed and conducted with the end thought of rapid translation into current LSD1 trials, data presented in this manuscript has the potential to significantly improve the survival outcomes for Ewing sarcoma patients as well as demonstrating to the scientific community how synergy experiments should be conducted. The data is technically sound with multiple replicates and each conclusion is supported by the data presented.

Minor comments for consideration

Introduction

1) As synergism with SP2509 was the main premise for this study, the authors should explain why SP2509 was chosen over other LSD1 inhibitors (GSK2879552 and ORY-1001). It would be helpful to state in the introduction that SP2509 is currently in Phase I clinical testing for Ewing sarcoma patients (NCT03600649) and that Ewing sarcoma cell lines are resistant to reversible inhibitors (Romo‐Morales et al., Pediatric Blood & Cancer, 2019).

2) To prevent confusion, it should be noted in the text that SP2509 was formally known as HCI2509.

Results section

1) Minor concern for this study was the decision to use Romidepsin which is only clinically approved for T-cell lymphoma. Considering the Phase 2 clinical trial of Romidepsin (NCT00112463) failed in 40 patients with metastatic or unresectable sarcoma, what is the likely hood of this HDAC inhibitor being chosen for further study in solid tumors?

2) Reference is required for the following statement “HDAC1 and HDAC2 inhibitor romidepsin and the reversible LSD1 inhibitor SP2509, were selected based on their respective targets in the NuRD complex” Lines 171-172.

3) Figure 1: Viability time point should be listed in the figure legend ie 24 or 72hr treatment. Image quality is also quite poor and should be corrected.

4) The authors state on Line 200 that “SP2509 showed synergy with all drugs apart from vincristine”. Can the authors postulate why? Is anything known for the cell lines in terms of previous chemotherapy treatment and does this explain sensitivity/resistance to the agents tested?

5) On lines 285-286, the authors elude to sensitivity of drugs may “simply be due to differences in doubling times”. This is quite possible considering the doubling times for the majority of cell lines ranges from 21-25hrs except for RDES (60hrs) (May et al., PLOS ONE, 2013). The authors should include doubling rates for each cell line in Table 1.

6) Is there any correlation between sensitivity to SP2509 and the other chemotherapeutic agents tested?

7) With all the different dosing schedules and agents, it was quite difficult throughout the manuscript to track which agents were synergistic over multiple cell lines. It would be nice to present a final table/figure summarizing which agents where synergistic across all four Ewing sarcoma cell lines and dosing schedules.

8) Minor notes, gene names should be in italics eg Line 275 TP53 and STAG2.

Discussion

1) Do the authors know whether SP2509 and SP2577 have similar IC50 values? This will have implications as to whether their synergistic findings can be replicated with SP2577 in clinical settings.

2) Although not used for clinical settings in Ewing sarcoma, SP2509 has been shown to synergise with docetaxel in prostate cancer (Gupta et al., 2016). This study should be referenced.

3) The dosing model presented in Supplementary Figure 2 is quite interesting and should be moved to the main text.

Reviewer #2: This study describes in vitro 2D analyses of cell viability for drug combination indices of an HDAC and an LSD1 inhibitor combined with standard chemotherapies used for treatment of ewing sarcomas. Overall the experiments appear well executed and described.

Although this is clearly not a mechanistic paper, the authors should better consider and discuss

i. the reasons supporting their hypothesis “that agents that interact with components of the NuRD complex would work synergistically with agents already utilized in the treatment of ES”.

ii. the rationale behind the choice of HDAC and LSD1 inhibitors including recent comments about how SP2509 may work (PMID: 29205263 PMID:31207107)

iii. possible mechanistic explanations for the results of the most and least synergistic combinations in different cell lines. For example it has been published that cells with low EWS-FLI1 expression are less proliferative PMID:28135250. Does this impact on the sensitivity of the agents used? The mechanism of action of some chemotherapeutic agents used depends on rapid cell cycling, e.g. SN-38. Can this be considered in light of A673’s relative insensitivity to topoisomerases? comments/considerations around reversal of insensitivity of

iv. the importance of scheduling. They only do one sequential experiment in which the order of drugs is changed.

v. the importance around time of treatment. Activity is reported at 72hours, however drugs targeting epigenetic modifying enzymes have been shown to benefit from prolonged exposure (EZH2 inhibitors in lymphoma). Longer-term assays could be useful to achieve maximal drug efficacy and therefore give a broader overview of their activity in combination with chemotherapeutic agents.

vi. Given the previous effects of LSD1 on migration (PMID:18381423) and the potential clinical significance of this phenotype, why was this not assessed or discussed?

Minor comments:

Page 3 line 44 sentence unclear – most commonly t(11;22)(q24;12) between the amino … – “12” should be q12 is missing and that fuses would be better than “between”

6. PLOS authors have the option to publish the peer review history of their article (what does this mean?). If published, this will include your full peer review and any attached files.

Reviewer #1: No

Reviewer #2: No

---

## [Author Response · Author response to Decision Letter 0]

8 Aug 2019

August 8, 2019

Dear Editors:

Please find our revised manuscript, “Small Molecule Inhibition of Lysine-Specific Demethylase 1 (LSD1) Alone and in Combination in Ewing Sarcoma Cell Lines” by Darcy Welch and Elliot Kahen et al., which we would like to submit for consideration of publication as an original research article in PLOS One.

 Thank you for the favorable review. We would indeed like to accept the offer of having this considered for the Targeted Anticancer Therapies and Precision Medicine Call for Papers as we agree it is within the scope of this collection.

 Also to clarify, there is no Sunshine Lab, LLC or commercial company involved in this research. All work done, and all authors are employees of an Academic Cancer Center. 

Below is a point by point response to the Reviewers’ comments.

Thank you for receiving our revised manuscript and considering it for review. We appreciate your time and look forward to your response.

Damon Reed, M.D., 

Associate Member, Department of Interdisciplinary Cancer Management and Sarcoma Departments

Director of Adolescent and Young Adult Program

Moffitt Cancer Center and Research Institute

12902 Magnolia Drive, Tampa, Florida 33612. 

Phone: 813-745-3242, Fax: 813-745-8337

Email: damon.reed@moffitt.org

Editor Comments:

-We have ensured style requirement compliance with this version.

2. Please provide additional information about each of the cell lines used in this work, including history, culture conditions and any quality control testing procedures (authentication, characterisation, and mycoplasma testing). For more information, please see http://journals.plos.org/plosone/s/submission-guidelines#loc-cell-lines.

-We have added this detail to the methods section.

3. Our staff editors have determined that your manuscript is likely within the scope of our Targeted Anticancer Therapies and Precision Medicine Call for Papers. This editorial initiative is headed by a team of Guest Editors for PLOS ONE: Andrew Cherniack, Anette Duensing, Steven Gray, Sunil Krishnan, Chandan Kumar-Sinha and Gayle Woloschak. The Collection will encompass a diverse range of research articles about the identification and classification of driver genes and somatic alterations, target and drug discovery, mechanisms of drug resistance, and early detection and screening. Additional information can be found on our announcement page: https://collections.plos.org/s/targeted-anticancer-therapies.

If you would like your manuscript to be considered for this collection, please let us know in your cover letter and we will ensure that your paper is treated as if you were responding to this call. If you would prefer to remove your manuscript from collection consideration, please specify this in the cover letter.

-We have noted above that we would like this article to be considered for this call for papers. Thank you for this opportunity.

4. Thank you for stating the following in the Financial Disclosure section: "This study was generously supported by the National Pediatric Cancer Foundation (www.nationalpcf.org). This work has been supported in part by the Translational Research Core at the H. Lee Moffitt Cancer Center & Research Institute, a NCI designated Comprehensive Cancer Center (P30-CA076292)."

We note that one or more of the authors are employed by a commercial company: 'Sunshine Lab LLC'.

-Sorry for the confusion. The Sunshine Lab is not an LLC, nor a commercial company. The Sunshine Lab is a term for the academic lab within Moffitt Cancer Center and all lab members are fully employed by Moffitt Cancer Center. It is analogous to naming a lab after a donor. The National Pediatric Cancer Foundation, a nonprofit and funder of the lab, does not have any say regarding the research performed in the lab.

Reviewer Comments:

Reviewer 1: Minor comments for consideration

Introduction

1) As synergism with SP2509 was the main premise for this study, the authors should explain why SP2509 was chosen over other LSD1 inhibitors (GSK2879552 and ORY-1001). It would be helpful to state in the introduction that SP2509 is currently in Phase I clinical testing for Ewing sarcoma patients (NCT03600649) and that Ewing sarcoma cell lines are resistant to reversible inhibitors (Romo‐Morales et al., Pediatric Blood & Cancer, 2019).

-Thank you for this recent reference, new since our submission (Line 65-72). 

2) To prevent confusion, it should be noted in the text that SP2509 was formally known as HCI2509.

-Thank you and agree. We have clarified this.

Results section

1) Minor concern for this study was the decision to use Romidepsin which is only clinically approved for T-cell lymphoma. Considering the Phase 2 clinical trial of Romidepsin (NCT00112463) failed in 40 patients with metastatic or unresectable sarcoma, what is the likely hood of this HDAC inhibitor being chosen for further study in solid tumors?

-We agree that HDAC inhibitors have a poor track record in sarcomas when given at hematologic malignancy doses and schedules. Specific to this project, the NuRD complex may contain HDAC1 and/or 2 and we were both curious as to whether or not inhibition of HDACs would phenotypically copy LSD1 inhibition. We also anticipated the question of whether or not HDAC and LSD1 play similar roles in this complexes general function. We believe that our results show romidepsin is a promising agent for ES treatment but SP2509 is a better candidate due to less antagonism across agents. Additionally we agree with the pragmatic point of the reviewer that HDAC inhibitors are unlikely to be studied in sarcomas. Since thresholds of efficacy are much lower in vitro than in clinical trials, we wanted to explore if romidepsin had any activity worthy of further mechanistic studies. We modified the manuscript to include our reasoning in selecting romidepsin (Lines 74-77)

2) Reference is required for the following statement “HDAC1 and HDAC2 inhibitor romidepsin and the reversible LSD1 inhibitor SP2509, were selected based on their respective targets in the NuRD complex” Lines 171-172.

-We have added the reference as well as adding additional information on our reasoning for selection in the introduction (Line 197).

3) Figure 1: Viability time point should be listed in the figure legend ie 24 or 72hr treatment. Image quality is also quite poor and should be corrected.

- We are sorry about the final image quality which we’ve learned is due to the generated pdf having a process that turned 600dpi TIFF files into jpegs. We have found that in the upper right of each figure is a link that can be clicked to quickly download the figure in the high res which was indeed submitted and will be in the final publication. The figure legend has been amended with the appropriate 72hr treatment.

4) The authors state on Line 200 that “SP2509 showed synergy with all drugs apart from vincristine”. Can the authors postulate why? Is anything known for the cell lines in terms of previous chemotherapy treatment and does this explain sensitivity/resistance to the agents tested?

-We appreciate the suggestion to comment further on this finding. We have added a sentence in the discussion postulating why (Line 362-377). In terms of prior therapy for the cell lines, TC-71 was derived post-chemo from a biopsy of a locally recurrent tumor; TC-32 was derived post-chemo; A673 is unknown; RD-ES was derived from a primary osseous of the humerus. This info has been added to the manuscript and briefly discussed.

5) On lines 285-286, the authors elude to sensitivity of drugs may “simply be due to differences in doubling times”. This is quite possible considering the doubling times for the majority of cell lines ranges from 21-25hrs except for RDES (60hrs) (May et al., PLOS ONE, 2013). The authors should include doubling rates for each cell line in Table 1.

-Very helpful and we agree. We have added both the doubling time and the suggested reference.

6) Is there any correlation between sensitivity to SP2509 and the other chemotherapeutic agents tested?

- The strongest correlation occurs between SP2509 and DNA damaging agents. This has been added to the manuscript along with Figure 4 and a possible explanation of the mechanism (Lines 354-358).

7) With all the different dosing schedules and agents, it was quite difficult throughout the manuscript to track which agents were synergistic over multiple cell lines. It would be nice to present a final table/figure summarizing which agents where synergistic across all four Ewing sarcoma cell lines and dosing schedules.

-A table of the top concentrations across all 4 cell lines has been added to the supplementary tables (S3 Table).

8) Minor notes, gene names should be in italics eg Line 275 TP53 and STAG2.

-Thank you again, this has been fixed.

Discussion

1) Do the authors know whether SP2509 and SP2577 have similar IC50 values? This will have implications as to whether their synergistic findings can be replicated with SP2577 in clinical settings.

- Since SP2577 involves company collaboration and review and we have limits on being able to publish data with the clinical compound, we have not included this agent in this report. We indeed tested both SP2577 and SP2509. We found that the IC50’s were similar for both compounds. SP2577 was slightly higher but on the same order of magnitude as SP2509. The company did not provide any financial support nor did they have any say in which experiments were performed. 

2) Although not used for clinical settings in Ewing sarcoma, SP2509 has been shown to synergise with docetaxel in prostate cancer (Gupta et al., 2016). This study should be referenced.

-We have added this reference and comment (Lines 375-377).

3) The dosing model presented in Supplementary Figure 2 is quite interesting and should be moved to the main text.

-Thank you. We struggled with including this in the main text versus supplemental but are happy to include this in the main body of the manuscript. It is now Fig 5.

Reviewer #2: This study describes in vitro 2D analyses of cell viability for drug combination indices of an HDAC and an LSD1 inhibitor combined with standard chemotherapies used for treatment of ewing sarcomas. Overall the experiments appear well executed and described.

Although this is clearly not a mechanistic paper, the authors should better consider and discuss

-We agree that this is not a mechanistic paper and more of a phenotypic paper regarding responses to agents. We do thank the reviewer for these strong comments and we hope our responses strengthen the interpretation and contextualization of the data. We agree there are many directions to go in terms of assays, schedules and time points. There are always more experiments to do and questions than answers. We are not prepared to do additional order of addition assays, or other experiments and it does not appear that this is being asked by the reviewer. We do add comments with this revision to help get these good points into the discussion to suggest next steps and to update the citations with emerging data regarding LSD1 and Ewing sarcoma.

i. the reasons supporting their hypothesis “that agents that interact with components of the NuRD complex would work synergistically with agents already utilized in the treatment of ES”.

-We have added to the introduction that the fusion oncoprotein is considered to be the driver of Ewing sarcoma and that the rationale for targeting both the fusion protein activity along with agents known to provide clinical benefit seems like a promising next step in treating Ewing sarcoma. It is currently unclear how the standard agents affect the fusion protein function. 

ii. the rationale behind the choice of HDAC and LSD1 inhibitors including recent comments about how SP2509 may work (PMID: 29205263 PMID:31207107)

-We added comments and references in the introduction and discussion on our rationale in addition to reviewer #1’s comment and feel we have addressed this point up to current knowledge (Lines 65-77). 

iii. possible mechanistic explanations for the results of the most and least synergistic combinations in different cell lines. For example it has been published that cells with low EWS-FLI1 expression are less proliferative PMID:28135250. Does this impact on the sensitivity of the agents used? The mechanism of action of some chemotherapeutic agents used depends on rapid cell cycling, e.g. SN-38. Can this be considered in light of A673’s relative insensitivity to topoisomerases? comments/considerations around reversal of insensitivity of

-We thank the reviewer for this. The comment again we feel is addressed by responses above to reviewer #1’s comment. Franzetti et al. has been added to the manuscript and we agree the results are intriguing (Lines 363-370). 

iv. the importance of scheduling. They only do one sequential experiment in which the order of drugs is changed.

- Our initial experiment did not show significant differences in the order of addition so we decided not to investigate further. We comment that future studies exploring dose and schedule are warranted.

v. the importance around time of treatment. Activity is reported at 72hours, however drugs targeting epigenetic modifying enzymes have been shown to benefit from prolonged exposure (EZH2 inhibitors in lymphoma). Longer-term assays could be useful to achieve maximal drug efficacy and therefore give a broader overview of their activity in combination with chemotherapeutic agents.

-We added a comment as we do agree with this point (Line 415). 

vi. Given the previous effects of LSD1 on migration (PMID:18381423) and the potential clinical significance of this phenotype, why was this not assessed or discussed?

-While migration assays are employed at times, we are not sure how drug effects on the migratory phenotype are applicable in the context of the performed experiments and tend to favor cell viability as the endpoint for these first screens of activity. It is difficult to imagine clinical activity from an agent that affects cell migration over viability. While we do believe migration, environment, metastases are indeed aspects of tumor biology that are tremendously important, the scope of this project was to measure viability towards translation to clinic.

Minor comments:

Page 3 line 44 sentence unclear – most commonly t(11;22)(q24;12) between the amino … – “12” should be q12 is missing and that fuses would be better than “between”

-We have fixed this, thank you.

---

## [Decision Letter · Decision Letter 1]

26 Aug 2019

[EXSCINDED]

Small Molecule Inhibition of Lysine-Specific Demethylase 1 (LSD1) and Histone Deactylase (HDAC) Alone and in Combination in Ewing Sarcoma Cell Lines

PONE-D-19-18074R1

Dear Dr. Reed,

We are pleased to inform you that your manuscript has been judged scientifically suitable for publication and will be formally accepted for publication once it complies with all outstanding technical requirements.

With kind regards,

Arun Rishi, Ph.D.

Academic Editor

PLOS ONE

Additional Editor Comments (optional):

Reviewers' comments:

Reviewer's Responses to Questions

**Comments to the Author**

1. If the authors have adequately addressed your comments raised in a previous round of review and you feel that this manuscript is now acceptable for publication, you may indicate that here to bypass the “Comments to the Author” section, enter your conflict of interest statement in the “Confidential to Editor” section, and submit your "Accept" recommendation.

Reviewer #1: All comments have been addressed

Reviewer #2: All comments have been addressed

2. Is the manuscript technically sound, and do the data support the conclusions?

Reviewer #1: Yes

Reviewer #2: Yes

3. Has the statistical analysis been performed appropriately and rigorously? 

Reviewer #1: Yes

Reviewer #2: N/A

4. Have the authors made all data underlying the findings in their manuscript fully available?

Reviewer #1: Yes

Reviewer #2: Yes

5. Is the manuscript presented in an intelligible fashion and written in standard English?

Reviewer #1: Yes

Reviewer #2: Yes

6. Review Comments to the Author

Reviewer #1: (No Response)

Reviewer #2: (No Response)

7. PLOS authors have the option to publish the peer review history of their article (what does this mean?). If published, this will include your full peer review and any attached files.

Reviewer #1: No

Reviewer #2: No

---

## [Editor Report · Acceptance letter]

9 Sep 2019

PONE-D-19-18074R1 

Small molecule inhibition of lysine-specific demethylase 1 (LSD1) and histone deacetylase (HDAC) alone and in combination in Ewing sarcoma cell lines 

Dear Dr. Reed:

I am pleased to inform you that your manuscript has been deemed suitable for publication in PLOS ONE. Congratulations! Your manuscript is now with our production department. 

With kind regards,

on behalf of

Prof Arun Rishi 

Academic Editor

PLOS ONE